# GWAS of Reproductive Traits in Large White Pigs on Chip and Imputed Whole-Genome Sequencing Data

**DOI:** 10.3390/ijms232113338

**Published:** 2022-11-01

**Authors:** Xiaoqing Wang, Ligang Wang, Liangyu Shi, Pengfei Zhang, Yang Li, Mianyan Li, Jingjing Tian, Lixian Wang, Fuping Zhao

**Affiliations:** 1Key Laboratory of Animal Genetics, Breeding and Reproduction (Poultry) of Ministry of Agriculture and Rural Affairs, Institute of Animal Science, Chinese Academy of Agricultural Sciences, Beijing 100193, China; 2Laboratory of Genetic Breeding, Reproduction and Precision Livestock Farming, School of Animal Science and Nutritional Engineering, Wuhan Polytechnic University, Wuhan 430023, China

**Keywords:** genotype imputation, genome-wide association analysis, Large White pigs, total number born, number of stillborn, gestation length

## Abstract

Total number born (TNB), number of stillborn (NSB), and gestation length (GL) are economically important traits in pig production, and disentangling the molecular mechanisms associated with traits can provide valuable insights into their genetic structure. Genotype imputation can be used as a practical tool to improve the marker density of single-nucleotide polymorphism (SNP) chips based on sequence data, thereby dramatically improving the power of genome-wide association studies (GWAS). In this study, we applied Beagle software to impute the 50 K chip data to the whole-genome sequencing (WGS) data with average imputation accuracy (*R*^2^) of 0.876. The target pigs, 2655 Large White pigs introduced from Canadian and French lines, were genotyped by a GeneSeek Porcine 50K chip. The 30 Large White reference pigs were the key ancestral individuals sequenced by whole-genome resequencing. To avoid population stratification, we identified genetic variants associated with reproductive traits by performing within-population GWAS and cross-population meta-analyses with data before and after imputation. Finally, several genes were detected and regarded as potential candidate genes for each of the traits: for the TNB trait: *NOTCH2*, *KLF3*, *PLXDC2*, *NDUFV1*, *TLR10*, *CDC14A*, *EPC2*, *ORC4*, *ACVR2A*, and *GSC*; for the NSB trait: *NUB1*, *TGFBR3*, *ZDHHC14*, *FGF14*, *BAIAP2L1*, *EVI5*, *TAF1B,* and *BCAR3*; for the GL trait: *PPP2R2B*, *AMBP*, *MALRD1*, *HOXA11*, and *BICC1*. In conclusion, expanding the size of the reference population and finding an optimal imputation strategy to ensure that more loci are obtained for GWAS under high imputation accuracy will contribute to the identification of causal mutations in pig breeding.

## 1. Introduction

Reproductive traits, such as total number born (TNB), number of stillborn (NSB), and gestation length (GL), are economically important traits that directly affect the economic benefits of the pig industry. The heritabilities of the TNB and NBS traits are about 0.1, whereas the heritability of the GL trait is about 0.3 [1,2]. The TNB trait is often used as one of the key indicators to measure the overall profitability of pig production, and the NSB trait is the most important feature to quantify the reproductive loss of pigs [3]. A study found that the NSB trait positively correlated with the TNB trait [4]. Several studies have shown that the key time for piglets to grow and mature is during late gestation [5], and a gestation length greater than 114 days may improve piglet survival after birth and can reduce postnatal mortality to a certain extent [6].

Genome-wide association studies (GWAS) have emerged as an efficient method for dissecting the genetic mechanisms of complex traits. Recently, GWAS have been widely employed to identify the candidate genes on traits in pigs, such as growth [7,8,9], reproduction [10,11,12], meat quality [13,14,15], and disease resistance [16,17]. The variant detection power of GWAS is affected by marker density. Now, GWAS are mainly carried out based on single-nucleotide polymorphism (SNP) chip data in animals. The existing commercial SNP chips are generally constructed based on information from a few breeds, which cannot capture the entire genetic variation of the genome due to ascertainment bias [18]. This poses difficulties in the identification of causal loci for economically important traits. Compared with SNP chip data, WGS data include all causal mutations. GWAS based on the WGS data can improve the power of the identification of mutations [19]. Although the cost of resequencing is decreasing, it is still expensive to resequence thousands of individuals with high coverage in animals. It is a more efficient approach to impute the SNP chip genotypic data to the WGS level, which is inexpensive [20].

Genotype imputation technology has been widely utilized in the processing of various genotype data in various fields and has become a particularly significant and routine tool. The accuracy of imputation is affected by many factors, such as reference population size, target population marker density, genetic distance between reference population and target populations, minor allele frequencies, and the imputation strategy [21,22,23]. The application of genotype imputation in GWAS can greatly narrow the genomic regions associated with target traits, which allows GWAS to find causal variants. Moreover, genotype imputation also provided the basis for the meta-analysis of GWAS. The use of imputed data for fine mapping led to the identification of many new key sites that increase the risk of type 2 diabetes in humans [24]. It was reported that the power of GWAS was dramatically improved using imputation-based WGS data, and out of 88 significant SNPs associated with the body shape of pigs, 85 were identified in imputation-based WGS data [25].

In this study, we selected 30 key ancestral individuals related to the target population for whole-genome resequencing, and then used them as a reference population to impute the 50K chip data to the WGS data. We identified genetic variants associated with the TNB, NSB, and GL traits in Large White pigs by performing GWAS with data before and after imputation. Therefore, the objective of this study was to impute the 50K chip data to the WGS data, and to analyze the possible factors influencing the imputation accuracy, so as to provide a certain reference for imputing the chip data to the WGS data. Furthermore, the data before and after imputation were used to conduct within-population GWAS and cross-population meta-analyses on the two different lines of Large White pigs. Our results provide some information for the breeding of important reproductive traits in pigs.

## 2. Results

### 2.1. Descriptive Statistics of Phenotypic Data

The descriptive statistics of the adjusted phenotypes for the TNB, NSB, and GL traits in the two Large White pig lines are shown in Table 1. The distribution of the phenotypes for the TNB, NSB, and GL traits is shown in Appendix A.

### 2.2. Genotype Imputation and Imputation Accuracy

We performed initial quality control of the WGS reference and target chip data. After quality control, 14,561,445 and 43,549 SNPs were retained in the 18 autosomes of the reference and target populations, respectively. We summarized the number of SNPs before and after imputation, the number of SNPs in imputation-based WGS data after quality control in each chromosome. In addition, we calculated the average imputation accuracy *R*^2^ before and after the quality control at *R*^2^ > 0.8. These are shown in Appendix A. Figure 1a,b represent the number of loci and imputation accuracy of the 18 autosomes before and after quality control, respectively. After imputation, 14,561,445 SNPs were obtained from the 18 autosomes of 2655 Large White pigs in this study, with 11,521,836 loci remaining after quality control according to *R*^2^ < 0.8 and MAF < 0.05. In addition, when we applied the same quality control condition to each line, 10,006,597 and 9,944,741 SNPs were retained after quality control for the Canadian and French lines, respectively. We calculated the average imputation accuracy *R*^2^ for all loci before and after quality control, which were 0.876 and 0.943, respectively.

### 2.3. Genome-Wide Association Studies

#### 2.3.1. GWAS for Data before Imputation

For the TNB trait, the Manhattan plots are shown in Figure 2a–d. The Q-Q plots are shown in Appendix A, with genome inflation factors between 0.950 and 1.000 (Appendix A). For the GWAS analysis within two lines of Large White pigs, no genome-wide significant SNPs were detected. Two SNPs (SSC4: 101,156,553 and SSC8: 30,016,379) at the suggestive significant level were observed in the Canadian line and only one SNP (SSC8: 56,076,247) at the suggestive significant level was observed in the French line. In the GWAS analysis of the combined two lines of Large White population, one SNP (SSC8: 56,076,247) at the genome-wide significant level was observed. There was one significant SNP (SSC8: 56,076,247) at the genome-wide level and one SNP (SSC5: 79,145,588) at the suggestive significant level in the cross-population meta-analyses (Table 2).

For the NSB trait, the Manhattan plots are shown in Figure 3a–d. The Q-Q plots are shown in Appendix A, with genome inflation factors between 0.957 and 1.006 (Appendix A). For the Canadian line, one genome-wide significant SNP (SSC10: 17,881,060) was detected, and four SNPs (SSC4: 123,733,425, SSC18: 6,400,611, SSC18: 5,909,015, and SSC1: 9,677,339) at the suggestive significant level were observed. For the French line, one genome-wide significant SNP (SSC6: 23,735,225) was detected, and one SNP (SSC14: 138,357,861) at the suggestive significant level was observed. In the GWAS analysis of the combined two lines of Large White population, five SNPs (SSC4: 125,301,443, SSC18: 5,909,015, SSC11: 70,923,126, SSC11: 70,551,880, and SSC1: 9,677,339) at the suggestive significant level were observed. The suggestive significant SNPs in the cross-population meta-analyses were the same as in the combined Large White population (Table 2).

For the GL trait, the Manhattan plots are shown in Figure 4a–d. Q-Q plots are shown in Appendix A, with genome inflation factors between 0.973 and 1.024 (Appendix A). For the GWAS analysis within both Large White lines, no genome-wide significant SNPs were detected. One SNP (SSC6: 156,647,853) at the suggestive significant level was observed in the Canadian line and two SNPs (SSC11: 25,293,190, SSC1: 254,755,615) at the suggestive significant level were observed in the French line. In the GWAS analysis of the combined lines of Large White population, one genome-wide significant SNP (SSC1: 254,755,615) and one SNP (SSC14: 61,937,863) at the suggestive significant level were detected. There were two SNPs (SSC1: 254,755,615 and SSC12: 3,251,323) at the suggestive significant level in the cross-population meta-analyses (Table 2).

#### 2.3.2. GWAS for Data after Imputation

For the TNB trait, the Manhattan plots are shown in Figure 2e–h. The Q-Q plots are shown in Appendix A, with genome inflation factors between 0.969 and 0.993 (Appendix A). For the GWAS analysis within both Large White lines, no genome-wide significant SNPs were detected. There were 147 SNPs at the suggestive significant level in the Canadian line and 136 SNPs at the suggestive significant level were in the French line. In the GWAS analysis of the combined lines of Large White population, there were 3 SNPs at the significant level and 203 SNPs at the suggestive significant level. There were 175 SNPs at the suggestive significant level in the cross-population meta-analyses (Table 3).

For the NSB trait, the Manhattan plots are shown in Figure 3e–h. The Q-Q plots are shown in Appendix A, with genome inflation factors between 0.938 and 1.036 (Appendix A). For the Canadian line, seven genome-wide significant SNPs and 708 suggestive significant SNPs were observed. For the French line, no genome-wide significant SNPs were detected and 152 SNPs at the suggestive significant level were observed. In the GWAS analysis of the combined lines of Large White population, one SNP at the genome-wide significant level and 385 suggestive significant SNPs were observed. One genome-wide significant SNP and 506 SNPs at the suggestive significant level were observed in the cross-population meta-analyses (Table 4).

For the GL trait, the Manhattan plots are shown in Figure 4e–h. The Q-Q plots are shown in Appendix A, with genome inflation factors between 0.975 and 1.024 (Appendix A). For the GWAS analysis within both Large White lines, no genome-wide significant SNPs were detected. Seven SNPs at the suggestive significant level were observed in the Canadian line and 136 SNPs at the suggestive significant level in the French line. In the GWAS analysis of the combined lines of the Large White population, no genome-wide significant SNPs and 156 SNPs at the suggestive significant level were detected. There were 48 SNPs at the suggestive significant level in the cross-population meta-analyses (Table 5).

### 2.4. Bioinformatics Annotation Analysis

In this study, GWAS based on 50K chip data and WGS data were used to detect candidate functional genes. According to the *Sus Scrofa* 11.1 pig genome, candidate genes were detected within a 20 kb region centering each significant and suggestive SNP.

For the TNB trait, 2 and 30 genes were found for 50K chip data and imputed WGS data, respectively. Additionally, one gene was simultaneously identified in both sets of data. For the NSB trait, 5 and 47 genes were found for 50K chip and imputed WGS data, respectively. Moreover, one gene was simultaneously identified in both datasets. For the GL trait, 3 and 14 genes were found for 50K chip and imputed WGS data, respectively. Furthermore, one gene was simultaneously identified in both datasets.

## 3. Discussion

### 3.1. Imputation of 50K Chip Data to WGS Data

In recent years, genotype imputation has been widely applied with the rapid decline in the cost of whole-genome resequencing data and the need for high-density markers. Genotype-population can be used to impute data with lower-density markers to the WGS data, and the imputation accuracy may be affected by the size of the reference population, the genetic distance between the reference and target populations, and the imputation strategy [23].

#### 3.1.1. Reference Population Size and Imputation Accuracy

The thirty Large White pigs used as the reference population in this study were the ancestral individuals in the population. The genetic distance between the reference and the target populations was relatively close. In addition, 14 pigs were also genotyped by 50K chip data and participated in the subsequent analysis. In this study, the accuracy of genotype imputation was higher than 0.858 for each chromosome before quality control, with an average imputation accuracy of 0.876 for 18 chromosomes, and higher than 0.928 for each chromosome after quality control, with an average imputation accuracy of 0.942. Quality control was applied in each population the loci loss rate of each population was 13.15% and 13.69% for the Canadian and French line-pigs, respectively. However, we imputed target pigs of the Canadian and French lines to WGS data by using 13 Canadian-line and 17 French-line Large White pigs as the reference population. The imputation accuracy before quality control was lower than that of the combined reference population. The imputation accuracy was 0.830 and 0.825 in Canadian and French lines, respectively. After quality control, the imputation accuracy was almost the same as that of the combined reference population, which was 0.943 and 0.944, respectively. However, the site loss rate after quality control was much higher than that of the combined reference population, which was 29.07% and 30.18%, respectively, being more than twice the loss rate of the combined reference group. The variation in the number of the reference population may be an important reason for this phenomenon. Using Beagle software, we imputed a medium-density chip of 50K to a high-density chip of 777K. As the size of reference population increased from 488 to 1229, their imputation error rate decreased from 0.67% to 0.41% [26]. We used Beagle software to impute the GBS data to the WGS data in Landrace pigs with 20 in the reference population and Large White pigs with 40 in the reference population, resulting in imputation accuracy of 0.42 and 0.45, respectively [27].

#### 3.1.2. Genetic Distance between Reference and Target Populations and Imputation Accuracy

The imputation accuracy of GBS data imputed to WGS data in the study on Large White pigs was 0.42 before quality control [27], which was much smaller than the 0.876 in this study before quality control. The possible reason was that the reference population in this study contained the key individuals in the population, and the genetic relationship between the target and the reference population was close. In a previous study, imputation was performed from 600K chip data to WGS data using multiple pig populations, and the average imputation accuracy before quality control was 0.49 [28]. In another study, Beagle software was used to impute the 60K chip data of 933 F2 populations to the WGS data. In this study, the 117 reference populations included 19 ancestors in F2 generations. The genotypic concordance and imputation accuracy were 0.89 and 0.80, respectively using cross-validation procedures [29]. In our study, we used 20-fold cross-validation procedures to evaluate the imputation genotypic concordance and imputation accuracy of chromosomes 1, 6, and 12; the genotypic concordance of the three chromosomes was 0.931, 0.936, and 0.899, respectively, and the imputation accuracy was 0.866, 0.867, and 0.812, respectively, which is close to a previously reported finding [29]. In a study on the imputation of multibreed sheep, if the individuals related to the individuals to be imputed were removed from the reference population, the concordance and accuracy of imputation reduced by 2.63% and 4.60%, respectively [30].

#### 3.1.3. Imputation Strategy and Imputation Accuracy

Researchers used Beagle software to impute 60K and 600K chip data to WGS data in a chicken population, obtaining an imputation accuracy of 0.620 and 0.812, respectively. In two-step imputation approach, the authors performed indirect imputation from 60K to 600K chip data and then from 600K chip to WGS data with an average imputing accuracy of 0.742 [22]. Researchers imputed 5K to 50K chip data and then from 50K chip data to HD data in sheep, which was superior to directly imputing 5K chip data to HD data, which increased the genotypic concordance by 5.67% [30]. In a study of genotype imputation in Holstein cattle, first imputing 50K chip data to the HD data and then to the WGS data improved the imputation accuracy from 0.28 to 0.65 compared with imputing 50K chip data directly to the WGS data, but was still lower than the imputation accuracy of 0.77 for imputing from the HD to the WGS data [31]. In a study of a small cattle population, endangered German Black Pied cattle, the accuracy of the two-step imputing method was found to be 92.1%, while the imputation accuracy of the one-step method was 93.2%. The author also analyzed the possible reason for this phenomenon and found that the intermediate reference level was a small population that is not abundant, which caused the incorrect imputing of the low-density chip to the medium–high-density in the first step [32]. In future study, we can try to add a medium- to high-density chip and try the two-step imputation method to compare the imputation accuracy, and the genotypic concordance of two-step imputing can be improved compared with the previous method.

### 3.2. Potential Candidate Genes

Imputing the chip data to WGS data using genotype imputation will allow more marker loci to be obtained for GWAS analysis at low cost. In this study, 50K chip data were imputed to WGS data, and the average imputation accuracy was 0.943 after quality control, so GWAS based on imputed WGS data were convincing. Compared with GWAS using the chip data, GWAS based on imputed WGS data detected more potential candidate genes. In addition, the meta-analysis improved the power of detection for SNPs by combining different populations. The advantage of meta-analyses has been reported in pigs [10,33,34]. In our study, we detected novel significant SNPs in the meta-analysis compared with single-breed analyses. However, there were no candidate genes within the 20 kb region centering each novel significant SNPs in the meta-analysis.

For the TNB trait, a number of candidate genes located within 20 kb of genome-wide significant and suggestive significant SNPs were identified in both lines. Among them, the *NOTCH2* gene plays an important role in pregnancy recognition and corpus luteum maintenance in mice [35]. A study indicated that the *NOTCH2* gene can inhibit the synthesis of estradiol [36]. Another study showed a role of *NOTCH2* in T-cell differentiation in subsets of T cells between intrauterine growth-retarded groups and normal groups [37]. In the early stages of human embryogenesis showed, *KLF3* is a transcription factor that persists during the transition from the zygote to the morula stage [38]. The *KLF3* gene may regulate fatty acid use in the intestine and reproductive tissue [39]. *PLXDC2* may play a role in reproduction and ectopic pregnancies [40]. A study showed a role for the *TBX10* gene in embryo development and diseases of mice [41]. It has been revealed that maternal nutrition in sows may alter birth weight mainly by regulating placental lipid and energy metabolism, and the *NDUFV1* gene plays an important role in energy metabolism [42]. The expression level of *NDUFV1* was downregulated in the placenta tissues compared with the normal pregnancy group, and the *NDUFV1* gene is involved in energy production processes in the mitochondrial matrix and membrane [43]. The *TLR10* gene can be expressed in the endometrium, conceptus, and chorioallantoic tissues of pigs, which may play a key role in regulating mucosal immune responses to support the establishment and maintenance of pregnancy [44]. The *CDC14A* gene can regulate oocyte maturation in mice [45]. The *CDC14A* gene is a possible candidate gene for protein yield associated with milk production in North American Holstein cattle [46]. The *CDC14A* gene is a candidate gene for body size traits in pigs [25]. The *EPC2* gene was found to be a novel candidate gene associated with reproductive performance in indigenous Chinese pigs [47]. The *ORC4* gene plays an important role in polar body extrusion during oogenesis [48,49,50]. The *ACVR2A* gene is widely expressed in ovarian granulosa cells and closely related to granulosa cell proliferation and follicular development [51]. The *ACVR2A* gene is a candidate gene for reproductive traits in pigs [52]. A study showed that *ACVR2A* is associated with female fertility in Japanese Black cattle [53]. The *GSC* gene can be used as an early marker of embryonic differentiation and describe embryonic diversity in pigs [54].

For the NSB trait, a number of candidate genes located within 20 kb of the genome-wide significant and suggestive significant SNPs were identified in both lines. Among them, the *NUB1* gene has been reported to be associated with milk production traits in cows and sheep [55,56]. The *TGFBR3* gene was also reported to be associated with oocyte maturation in pigs [57]. The *ZDHHC14* gene may act as a marker and target for the clinical diagnosis and treatment of pre-eclampsia [58]. The *FGF14* gene may be a promising candidate gene associated with litter traits in pigs [59] and a potential candidate gene for teat number trait in Duroc pigs [60]. The *BAIAP2L1* gene may serve as a biomarker in ovarian cancer [61]. The *BCAR3* gene may provide new insights into the mechanism of local estrogen action in endometriosis [62], and may contribute to the complex tumor heterogeneity of ovarian cancer cells [63]. The *EVI5* gene displayed significantly differential expression in trophectoderm biopsies associated with live birth and no-implanting [64]. A study observed that the absence of the *TAF1B* gene in germline cells leads to the accumulation of late stage egg chambers in the ovaries [65]. The *TAF1B* gene is a candidate gene for congenital splay leg. Porcine splay leg syndrome is still one of the most important causes of piglet loss, which can be caused by myofibrillar hypoplasia [66].

For the GL trait, a number of candidate genes located within 20 kb of genome-wide significant and suggestive significant SNPs were identified in both lines. Among them, the *PPP2R2B* gene had a genetic significant effect on milk production traits in Chinese Holstein [67]. This gene may be associated with sperm motility in Duroc pigs as a candidate gene [68]. The *PPP2R2B* gene may act as an important reproductive driver gene [69]. A study found that the *AMBP* gene was overexpressed in the amniotic fluid of women without intra-amniotic infection/inflammation [70]. Increased concentrations of this *AMBP* gene are often considered an indicator of pre-eclampsia [71]. The *MALRD1* gene is associated with endometriosis in humans [72]. The *BICC1* gene is differentially expressed during prenatal development of skeletal muscle in Pietrain and Duroc pigs [73]. A study identified the *BICC1* gene as an important candidate gene of reproductive traits in Duroc pigs [74]. The *HOXA3*, *HOXA7*, *HOXA10*, and *HOXA11* genes were found to be candidates for reproductive traits in a study of runs of homozygosity in Jinhua pig [75]. It has also been shown that the *HOXA11* gene is expressed in the endometrium [76] and is associated with endometrial epithelial function [77].

## 4. Materials and Methods

### 4.1. Animals and Phenotype

The Large White pigs used in this study were from a commercial pig company in Shanghai, China, which were introduced from Canadian and French lines. Feeding and performance testing for these two lines were conducted on two different farms, with essentially the same level of nutritional management. A total of 13,379 reproduction records of 2655 individuals from parity 1 to 7 were collected during the period of 2014–2020, of which 1403 were from the Canadian line and 1252 were from the French line. According to pedigree information, there was no genetic connectedness between the two lines. Three reproductive traits, TNB, NSB, and GL, were selected for subsequent analysis. The DMUAI procedure of DMU software(Version 6, release 5.2) was used to adjust phenotype on the repeated records of multiple parities based on the pedigree information [78]. The statistical model is described below:$$y_{ijklm}=\mu +L_{i}+T_{j}+YS_{k}+a_{ijkl}+pe_{ijklm}+e_{ijklm}$$
where $y_{ijklm}$ is the phenotype, such as TNB, NSB, and GL traits; μ is the total mean; $L_{i}$ is the line effect; $T_{j}$ is the parity effect; $YS_{k}$ is the measured year-season effect, where the season is divided according to the month and consists of four levels (spring = March to May; summer = June to August; autumn = September to November; winter = December to February); and $a_{ijkl}$ is the additive genetic effect, with $a_{\;}$~*N*(0,$A\sigma_{a}^{2}$), where $\sigma_{a}^{2}$ is the additive genetic variance, A is the numerator relationship matrix, pe is the permanent environmental effect, with pe~*N*(0,$I\sigma_{pe}^{2}$), and $e_{ijklm}$ represents residuals.

### 4.2. SNP Chip Data

We selected 2655 Large White pigs as the target population. Genotyping was performed using a GeneSeek Porcine 50K array. The chip was designed according to Sus *Scrofa* 10.2 and contained 50,915 SNPs. We mapped autosomal SNPs to the latest version of the pig genome *Sus Scrofa* 11.1, resulting in 46,311 autosomal SNPs for subsequent analysis.

Quality control was performed by PLINK v1.90 software [79]. In each population, pigs with an individual call rate of lower than 0.9 were excluded. SNP call rates less than 0.9 were removed and we retained SNPs with minor allele frequencies (MAF) of 0.05 or higher. After quality control, 43,549 SNPs remained in the combined Large White population for subsequent analysis. For the Canadian and French lines of Large White pigs, we used 41,039 and 40,495 autosomal SNPs for subsequent analysis, respectively.

### 4.3. Reference Sequence Data

Based on the pedigree information, we first ranked the individuals in the Large White population according to the number of offspring. Then, we select the top thirty ancestral individuals that we called the key individuals in the Large White population as the reference population. Among these 30 Large White pigs, there were 13 Canadian-line pigs and 17 French-line pigs. In addition, fourteen pigs also had chip data and participated in the subsequent analysis. The whole-genome resequencing of 30 Large White pigs was carried out on an Illumina HiSeq platform with average sequencing depth of 10-fold. The initial quality of resequencing data was determined by Trimmomatic (version 0.39) [80]. The clean reads were mapped to the *Sus Scrofa* 11.1 reference sequence with BWA (version 0.7.17) software [81]. Afterwards, GATK (version 4.1.8.1) software was used to realign the mapped reads and call the SNPs [82]. A total of 21,039,605 SNPs were called by GATK. Quality control was performed by removing duplication sites and SNPs with no position information or located on sex chromosomes. We retained the SNPs with minor allele frequency (MAF) > 0.05, SNPs call rate > 0.9, and Hardy–Weinberg equilibrium (HWE) < 1.0 × 10^−6^; quality control was performed with VCFtools (version 0.1.16) [83]. After quality control, a total of 14,561,445 SNPs remained.

### 4.4. Genotype Imputation

Using the WGS reference data of 30 Large White pigs, the GeneSeek Porcine 50K chip data of 2655 target Large White pigs were imputed to WGS data. Genotype imputation was conducted with Beagle (version 5.2.2) software [84]. After imputation, quality control was performed with BCFtools (version 1.8) software in each of the two lines [85]. In each population, imputation accuracies lower than 0.8 and minor allele frequencies (MAFs) of lower than 0.05 were excluded. The imputation accuracy *R*^2^ at each SNP was the squared correlation between the known true genotypes and the expected dosages [86].

### 4.5. Genome-Wide Association Studies

In this study, we used the sum of an estimated breeding value (EBV) and a residual of an individual as the adjusted phenotype to conduct GWAS. The single SNP regression models were independently performed on GeneSeek Porcine 50K chip data and imputed WGS data using GCTA (version 1.93.3beta) [87]. The statistical model is described below:y=Xb+Wg+e
where y is the vector of the adjusted phenotypes, such as TNB, NSB, and GL traits; b is the vector of fixed effects; there was no fixed effect in the within-population analysis, and the line effect was added as a fixed effect in the combined Large White population. g is a vector of the SNP effects; W and X correspond to the correlation matrix of b and g, respectively; e is the vector of residual effects, with e~*N*(0, $I\sigma_{e}^{2}$).

In addition, the cross-population meta-analysis based on default method was conducted with METAL software (version “2011-03-25”) [88]. The default method in the METAL software combines *p*-values across studies taking into account the sample size and direction of the effect.

For 50K chip data, the threshold values were determined by the Bonferroni correction method. The threshold *p*-values for genome-wide significance and suggestive were set to −log10 (0.05/SNPs) and −log10 (1/SNPs), respectively. For imputed WGS data, we used 5 × 10^−8^ as a genome-wide significance level, which was also applied in human GWAS [89]. We adopted 5 × 10^−6^ as the suggestive level. The Manhattan and quantile-quantile (QQ) plots were drawn with the R package “qqman” [90].

### 4.6. Bioinformatics Annotation Analysis

The bioinformatics database BioMart (http://www.ensembl.org/, accessed on 27 August 2022) was used to screen candidate genes located within significant and suggestive loci. We only considered genes located in the ±20 kb region around significant and suggestive SNPs.

## 5. Conclusions

In this study, we imputed 50K chip data to WGS data, with an average imputation accuracy of 0.876 before quality control and 0.943 after quality control (*R^2^* > 0.8 and MAF> 0.05). The imputed WGS data for GWAS is cost-effective, which can reduce the mapping noise. These results provide useful, new insights into the genetic variation and genes associated with TNB, NSB, and GL traits in different lines of Large White pigs. However, further studies are needed to determine the optimal imputation strategy from chip to WGS data. GWAS based on chip data and imputed WGS data were performed for three reproductive traits in the Canadian and French lines of Large White pigs. Finally, combining the results of GWAS and bioinformatics annotation analysis, *NOTCH2*, *KLF3*, *PLXDC2*, *NDUFV1*, *TLR10*, *CDC14A*, *EPC2*, *ORC4*, *ACVR2A*, and *GSC* genes were identified as potential candidate genes associated with the TNB trait; *NUB1*, *TGFBR3*, *ZDHHC14*, *FGF14*, *BAIAP2L1*, *EVI5*, *TAF1B*, and *BCAR3* were considered potential candidate genes related to the NSB trait; and *PPP2R2B*, *AMBP*, *MALRD1*, *HOXA11*, and *BICC1* were detected as potential candidate genes related to the GL trait in Large White pigs. In addition, the size of the reference population used in this study was small, and the detection power of GWAS analysis was weak. Subsequently, we can consider expanding the size of the reference population and adopting a further fine imputation strategy to discover causal mutations and validate these identified SNPs and genes.

## Figures and Tables

**Figure 1 ijms-23-13338-f001:**
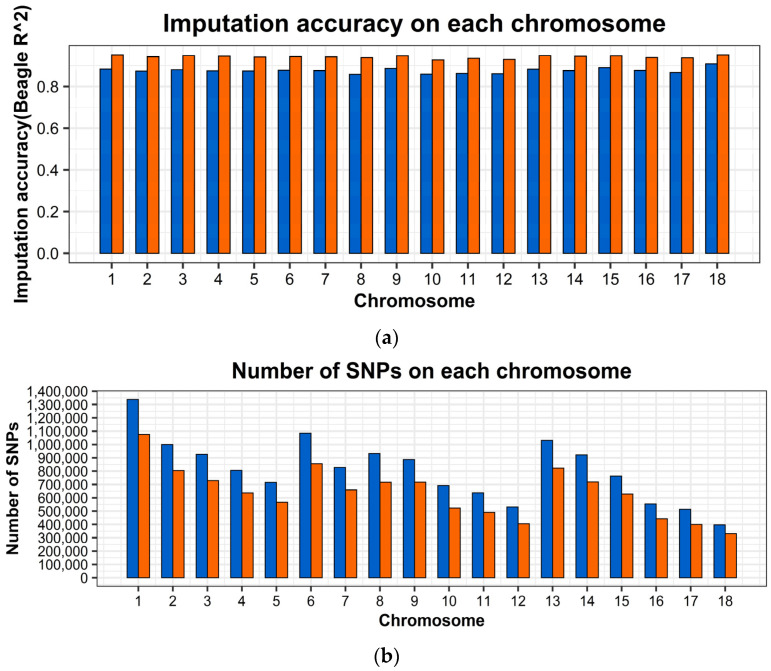
The summary of imputation accuracy and number of SNPs after imputation from 50K chip data to WGS data. (**a**) Imputation accuracy in each chromosome in Large White pigs. Imputation accuracy before quality control (blue), after quality control, with *R*^2^ > 0.8 (orange). (**b**) Number of SNPs after imputation with or without quality control in each chromosome in Large White pigs. Number of SNPs before quality control (blue), Number of SNPs after quality control (orange).

**Figure 2 ijms-23-13338-f002:**
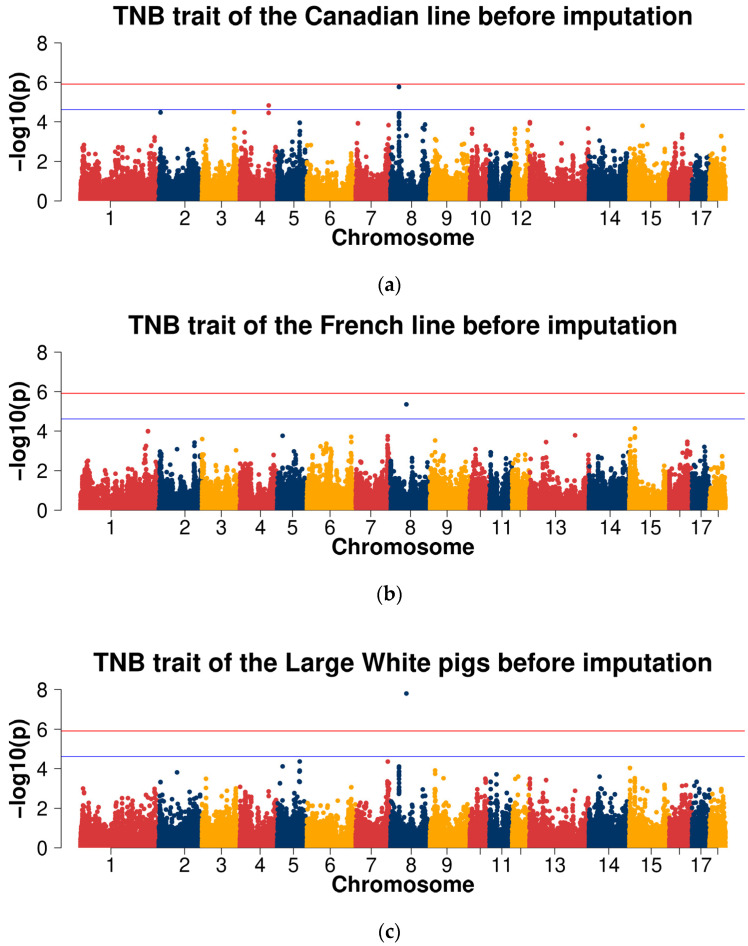
The Manhattan plots of association results of different populations for the total number born (TNB) trait using chip data (**a**–**d**) and imputed WGS data (**e**–**h**). (**a**,**e**) Canadian line; (**b**,**f**) French line; (**c**,**g**) combined lines of Large White pigs; (**d**,**h**) cross-population meta-analyses.

**Figure 3 ijms-23-13338-f003:**
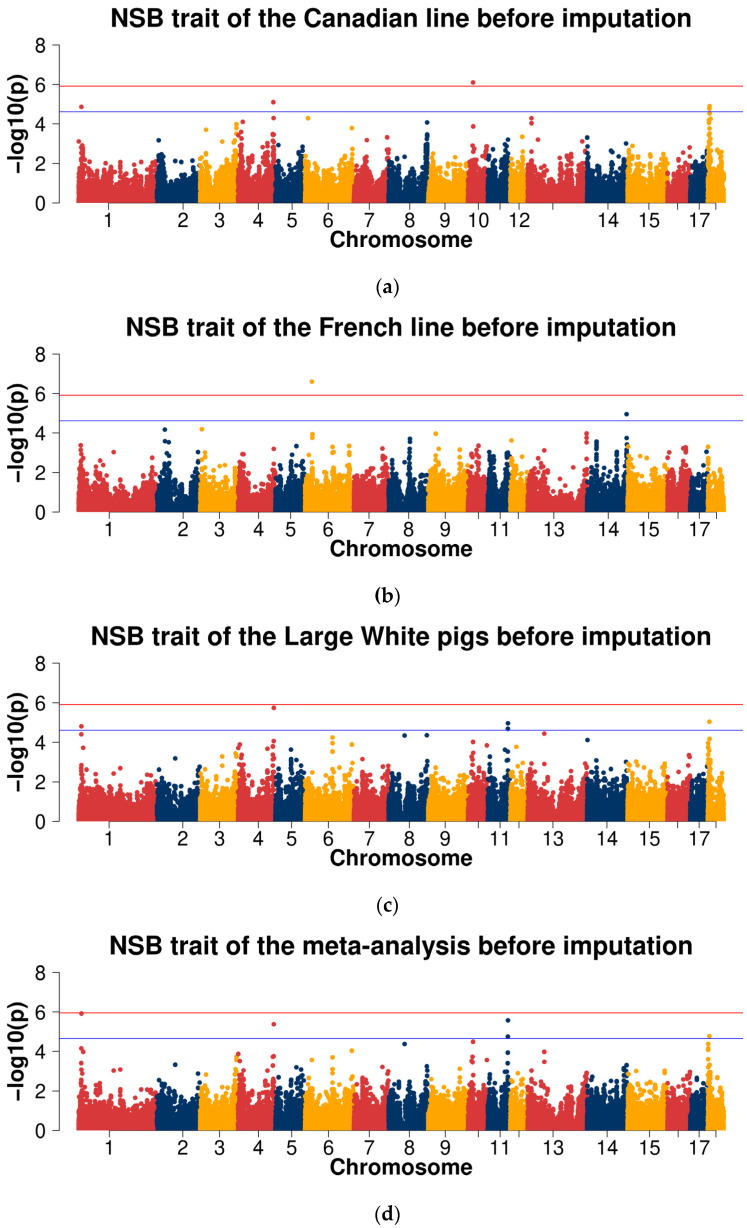
Manhattan plots of association results of different populations for the number of stillborn (NSB) trait using chip data (**a**–**d**) and imputed WGS data (**e**–**h**). (**a**,**e**) Canadian line; (**b**,**f**) French line; (**c**,**g**) combined lines of Large White pigs; (**d**,**h**) cross-population meta-analyses.

**Figure 4 ijms-23-13338-f004:**
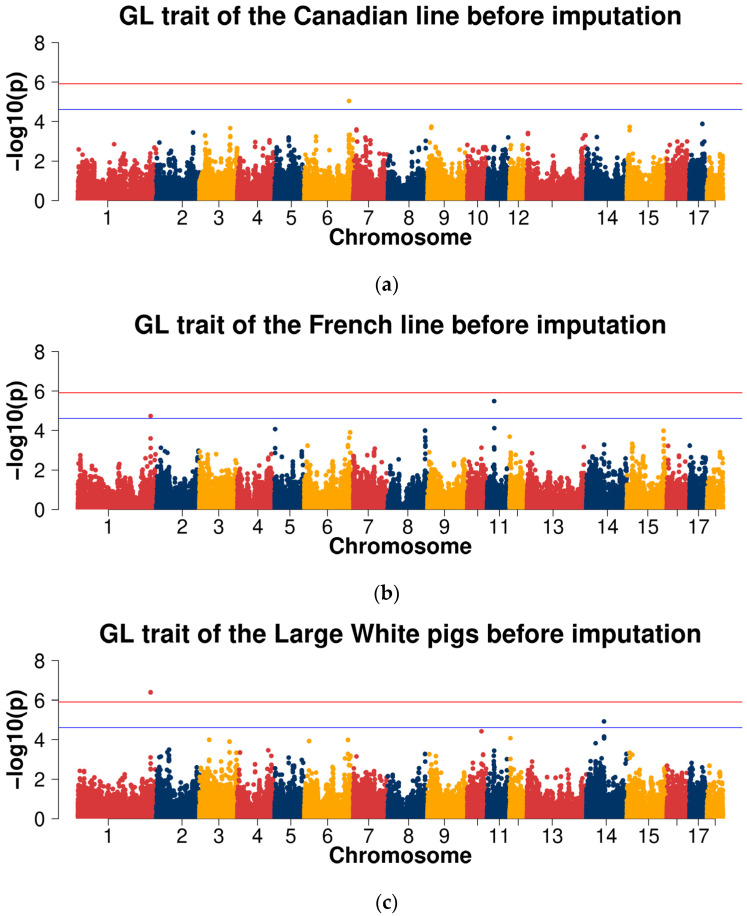
Manhattan plots of association results of different populations for the gestation length (GL) trait using chip data (**a**–**d**) and imputed WGS data (**e**–**h**). (**a**,**e**) Canadian line; (**b**,**f**) French line; (**c**,**g**) combined lines of Large White pigs; (**d**,**h**) cross-population meta-analyses.

**Table 1 ijms-23-13338-t001:** Descriptive statistics for total number born (TNB), number of stillborn (NSB), and gestation length (GL) in the two lines of Large White pigs.

Line	Trait	Samples Size	Mean	Standard Deviation	Minimum	Maximum
Canadian	TNB	1403	14.42	1.97	8.10	22.30
French		1252	14.29	1.72	7.85	21.03
Canadian	NSB	1403	1.26	0.85	−0.21	4.65
French		1252	1.16	0.74	−0.19	5.73
Canadian	GL	1403	114.59	0.93	111.35	117.69
French		1252	114.09	1.12	110.11	119.82

**Table 2 ijms-23-13338-t002:** The significant and suggestive SNPs in the genome with the total number born (TNB), number of stillborn (NSB), and gestation length (GL) traits using chip data in pigs.

Trait	SSC	SNP Name	SNP Position (bp)	*p* (Canadian)	*p* (French)	*p* (Combined LW)	*p* (Meta)	Candidate Gene
TNB	4	WU_10.2_4_111043929	101,156,553	1.47 × 10^−5^	6.50 × 10^−1^	2.39 × 10^−3^	6.20 × 10^−4^	*NOTCH2*
	5	ALGA0122851	79,145,588	1.11 × 10^−4^	3.71 × 10^−2^	4.26 × 10^−5^	2.22 × 10^−5^	-
	8	H3GA0024679	30,016,379	1.69 × 10^−6^	4.62 × 10^−1^	9.30 × 10^−5^	6.75 × 10^−5^	** *KLF3* **
	8	Affx-114981021	56,076,247	4.95 × 10^−4^	4.44 × 10^−6^	1.61 × 10^−8^	1.30 × 10^−8^	-
NSB	1	INRA0000573	9,677,339	1.37 × 10^−5^	1.35 × 10^−2^	1.54 × 10^−5^	1.20 × 10^−6^	*TMEM242*
	4	ALGA0029239	123,733,425	7.90 × 10^−6^	7.47 × 10^−1^	6.98 × 10^−4^	2.46 × 10^−3^	*FNBP1L*
	4	WU_10.2_4_136884741	125,301,443	1.94 × 10^−3^	6.29 × 10^−4^	1.78 × 10^−6^	4.20 × 10^−6^	** *TGFBR3* **
	6	WU_10.2_6_21881195	23,735,225	8.94 × 10^−1^	2.51 × 10^−7^	1.92 × 10^−2^	2.72 × 10^−4^	-
	10	ASGA0046895	17,881,060	7.95 × 10^−7^	4.08 × 10^−1^	9.49 × 10^−5^	3.22 × 10^−5^	-
	11	WU_10.2_11_78220891	70,923,126	6.20 × 10^−4^	1.30 × 10^−3^	1.08 × 10^−5^	2.65 × 10^−6^	-
	11	ALGA0063609	70,551,880	5.29 × 10^−3^	9.79 × 10^−4^	2.05 × 10^−5^	1.77 × 10^−5^	-
	14	MARC0062790	138,357,861	5.25 × 10^−1^	1.10 × 10^−5^	1.39 × 10^−2^	4.98 × 10^−4^	-
	18	WU_10.2_18_6267059	5,909,015	1.71 × 10^−5^	-	8.92 × 10^−6^	1.71 × 10^−5^	*NUB1*
	18	MARC0056921	6,400,611	1.26 × 10^−5^	6.43 × 10^−1^	6.66 × 10^−5^	4.79 × 10^−4^	*GIMAP2*
GL	1	ASGA0104591	254,755,615	1.82 × 10^−2^	1.84 × 10^−5^	4.08 × 10^−7^	3.58 × 10^−6^	*COL27A1*
	6	WU_10.2_6_144601434	156,647,853	8.97 × 10^−6^	1.21 × 10^−1^	2.60 × 10^−1^	3.05 × 10^−2^	-
	11	ALGA0124549	25,293,190	9.56 × 10^−1^	3.24 × 10^−6^	3.60 × 10^−4^	1.26 × 10^−3^	*VWA8*
	12	WU_10.2_12_3290782	3,251,323	2.42 × 10^−3^	2.74 × 10^−3^	8.38 × 10^−5^	2.03 × 10^−5^	*-*
	14	MARC0035949	61,937,863	4.86 × 10^−2^	1.48 × 10^−3^	1.18 × 10^−5^	2.98 × 10^−4^	** *BICC1* **

SSC, *Sus scrofa* chromosome. SNP name, name of significant and suggestive SNPs; SNP position (bp), the position of significant and suggestive SNPs. *p* (Canadian), *p*-value from within-population GWAS in the Canadian line. *p* (French), *p*-value from within-population GWAS in the French line. *p* (Combined LW), *p*-value from within-population GWAS in the combined two lines of Large White pigs. *p* (Meta), *p*-value from cross-population meta-analyses. Bolded text shows the potential candidate gene detected in both chip and WGS data.

**Table 3 ijms-23-13338-t003:** The genome significant and suggestive SNPs with the total number born (TNB) trait using imputed WGS data in pigs.

SSC	SNP_R (Mb)	T_SNP_P (bp)	SNP_N	*p* (Canadian)	*p* (French)	*p* (Combined LW)	*p* (Meta)	Candidate Gene
1	261.86–261.90	261,882,219	4	4.42 × 10^−1^	7.79 × 10^−7^	2.41 × 10^−2^	4.60 × 10^−3^	*TTLL11*
2	4.90–4.94	4,917,216	1	6.89 × 10^−7^	-	3.47 × 10^−6^	6.89 × 10^−7^	*UNC93B1/ALDH3B2*
2	4.94–4.98	4,964,340	19	5.72 × 10^−7^	6.61 × 10^−1^	2.02 × 10^−4^	8.56 × 10^−4^	*TBX10/NDUFV1*
2	127.88–127.92	127,904,283	13	4.21 × 10^−6^	-	8.20 × 10^−5^	4.21 × 10^−6^	-
3	6.81–6.85	117,058,450	1	3.78 × 10^−4^	-	4.61 × 10^−6^	3.78 × 10^−4^	-
3	117.04–117.08	6,831,554	1	6.12 × 10^−4^	1.65 × 10^−3^	1.08 × 10^−5^	3.29 × 10^−6^	-
4	117.63–117.67	117,647,397	1	9.90 × 10^−1^	4.47× 10^−6^	2.16 × 10^−3^	1.68 × 10^−3^	*CDC14A*
5	78.38–78.42	78,396,126	1	8.21 × 10^−5^	9.81 × 10^−3^	7.00 × 10^−6^	3.55 × 10^−6^	*COL2A1*
5	78.40–78.44	78,424,002	1	4.34 × 10^−6^	3.17 × 10^−3^	1.08 × 10^−7^	8.03 × 10^−8^	*SENP1*
5	78.53–78.57	78,554,745	1	1.14 × 10^−5^	3.20 × 10^−3^	2.80 × 10^−7^	1.84 × 10^−7^	*CCDC184*
5	79.02–79.12	79,102,021	11	1.30 × 10^−6^	2.89 × 10^−2^	1.13 × 10^−6^	5.21 × 10^−7^	-
5	79.12–79.16	79,144,763	8	6.11 × 10^−6^	3.86 × 10^−2^	2.66 × 10^−6^	2.50 × 10^−6^	-
5	79.24–79.28	79,262,197	2	1.07 × 10^−6^	7.45 × 10^−3^	8.06 × 10^−6^	3.25 × 10^−6^	*KANSL2*
5	79.24–79.29	79,271,540	7	1.38 × 10^−6^	2.44 × 10^−2^	5.50 × 10^−7^	4.31 × 10^−7^	*KANSL2*, *SNORA2C*
5	79.25–79.29	79,266,042	3	1.32 × 10^−5^	2.44 × 10^−2^	3.71 × 10^−6^	2.44 × 10^−6^	*KANSL2*, *SNORA2C*
5	79.70–79.74	79,716,662	1	4.02 × 10^−7^	4.12 × 10^−2^	1.97 × 10^−7^	3.66 × 10^−7^	
5	79.71–79.75	79,732,010	3	2.71 × 10^−7^	6.65 × 10^−1^	4.30 × 10^−4^	5.45 × 10^−5^	*SLC41A2*
7	115.99–116.04	116,017,230	9	2.94 × 10^−5^	1.30 × 10^−3^	5.09 × 10^−8^	1.55 × 10^−7^	*GSC*
7	115.06–116.10	116,082,776	2	1.44 × 10^−5^	2.58 × 10^−3^	3.67 × 10^−8^	1.76 × 10^−7^	*GSC*
7	116.32–116.36	116,344,726	1	1.62 × 10^−5^	-	1.63 × 10^−6^	1.62 × 10^−5^	-
8	29.96–30.00	29,982,306	2	5.03 × 10^−7^	-	2.49 × 10^−5^	5.03 × 10^−7^	-
8	30.04–31.11	31,089,176	87	4.23 × 10^−6^	1.79 × 10^−1^	1.91 × 10^−3^	1.98 × 10^−5^	***KLF3***, *FAM114A1, TLR10, TLR1, TLR6, WDR19, PDS5A*
8	56.06–56.10	56,076,247	1	5.72 × 10^−4^	5.79 × 10^−6^	2.33 × 10^−8^	1.94 × 10^−8^	-
10	54.58–54.62	54,600,820	1	3.16 × 10^−6^	1.11 × 10^−1^	5.38 × 10^−6^	7.38 × 10^−6^	*PLXDC2*
10	55.49-55.53	55,514,613	9	1.57 × 10^−4^	1.66 × 10^−2^	4.62 × 10^−6^	1.12 × 10^−5^	-
13	205.13–205.17	205,150,403	1	2.16 × 10^−6^	7.41 × 10^−1^	1.53 × 10^−4^	2.42 × 10^−4^	*RIPK4*
15	2.20–2.28	2,257,349	4	-	3.69 × 10^−5^	4.41 × 10^−6^	3.69 × 10^−5^	-
15	2.24–2.35	2,325,462	4	5.96 × 10^−3^	3.20× 10^−5^	3.10 × 10^−6^	2.46 × 10^−6^	-
15	2.50–2.56	2,538,624	5	-	1.41 × 10^−7^	2.79 × 10^−7^	1.41× 10^−7^	*MMADHC*
15	2.54–2.60	2,578,548	6	1.56 × 10^−1^	3.42 × 10^−7^	1.96 × 10^−6^	2.10 × 10^−5^	-
15	2.57–2.61	2,586,649	2	8.65 × 10^−1^	8.25 × 10^−7^	9.70 × 10^−5^	4.51 × 10^−4^	*LYPD6*
15	2.54–2.61	2,564,841	83	-	9.22 × 10^−7^	1.67 × 10^−6^	9.22 × 10^−7^	*LYPD6*
15	2.65–2.69	2,671,075	1	2.53 × 10^−3^	3.27 × 10^−4^	6.97 × 10^−6^	3.13 × 10^−6^	*LYPD6*
15	2.76–2.80	2,778,681	1	-	3.07 × 10^−6^	3.83 × 10^−5^	3.07 × 10^−6^	*LYPD6*
15	2.68–2.81	2,779,384	11	9.87 × 10^−1^	2,779,3842.99 × 10^−6^	5.85 × 10^−4^	1.39 × 10^−3^	*LYPD6*
15	2.90–2.95	2,933,736	6	-	7.06 × 10^−6^	3.30 × 10^−6^	5.68 × 10^−6^	*LYPD6B*
15	2.92–2.96	2,938,800	2	-	3.47 × 10^−6^	2.44 × 10^−6^	3.47 × 10^−6^	*LYPD6B*
15	3.24–3.28	3,264,770	1	4.51 × 10^−1^	4.48 × 10^−6^	6.54 × 10^−5^	2.17 × 10^−4^	*KIF5C*
15	3.26–3.30	3,278,046	2	9.74 × 10^−2^	1.03 × 10^−5^	3.90 × 10^−6^	2.29× 10^−5^	*KIF5C*
15	3.36–3.40	3,376,487	8	4.19 × 10^−3^	1.30 × 10^−4^	1.95 × 10^−6^	2.48 × 10^−6^	*KIF5C*
15	3.37–3.41	3,392,704	8	1.18 × 10^−3^	2.91 × 10^−3^	2.42 × 10^−6^	1.07 × 10^−5^	-
15	3.37–3.57	3,545,949	8	1.11 × 10^−2^	1.33 × 10^−5^	2.73 × 10^−7^	1.32 × 10^−6^	*EPC2*
15	4.15–4.43	4,408,571	14	2.42 × 10^−2^	4.58× 10^−5^	3.03 × 10^−6^	9.11 × 10^−6^	*ORC4*, *ACVR2A*
15	4.79–4.88	4,857,043	20	3.63 × 10^−1^	1.07 × 10^−6^	5.46 × 10^−3^	7.16 × 10^−3^	-

SSC, *Sus scrofa* chromosome. SNP_R, range of significant and suggestive SNPs region. SNP_N, number of significant and suggestive SNPs. T_SNP_P, the position (bp) of the top SNP in range of significant and suggestive SNPs region. *p* (Canadian), *p*-value from within-population GWAS in the Canadian line. *p* (French), *p*-value from within-population GWAS in the French line. *p* (Combined LW), *p*-value from within-population GWAS in the combined lines of Large White pigs. *p* (Meta), *p*-value from cross-population meta-analyses. The bolded text shows the potential candidate gene detected in both chip and WGS data.

**Table 4 ijms-23-13338-t004:** The genome significant and suggestive SNPs with the number of stillborn (NSB) trait using imputed WGS data in pigs.

SSC	SNP_R (Mb)	T_SNP_P (bp)	SNP_N	*p* (Canadian)	*p* (French)	*p* (Combined LW)	*p* (Meta)	Candidate Gene
1	9.04–9.42	9,404,407	12	3.60 × 10^−5^	1.11 × 10^−2^	1.07 × 10^−5^	2.06 × 10^−6^	*SYNJ2, ZDHHC14*
1	9.38–9.43	9,411,440	57	3.57 × 10^−5^	6.68 × 10^−4^	3.87 × 10^−7^	9.23 × 10^−8^	*ZDHHC14*
1	9.44–9.76	9,743,278	13	1.03 × 10^−4^	7.68 × 10^−3^	3.83 × 10^−5^	3.26 × 10^−6^	*ZDHHC14, TMEM242*
3	5.44–5.49	5,470,408	5	1.25 × 10^−2^	2.52 × 10^−5^	8.38 × 10^−6^	2.48 × 10^−6^	*TECPR1*, *BRI3*, *BAIAP2L1*
3	5.43–5.50	5,478,442	5	7.49 × 10^−3^	2.30 × 10^−5^	1.83 × 10^−6^	1.23 × 10^−6^	*TECPR1*, *BRI3*, *BAIAP2L1*
3	126.29–126.42	126,396,984	9	1.62 × 10^−6^	5.71 × 10^−1^	6.02 × 10^−5^	1.06 × 10^−4^	*CYS1*
3	126.38–126.43	126,405,359	6	6.22 × 10^−6^	4.31 × 10^−2^	3.09 × 10^−6^	2.96 × 10^−6^	*CYS1, KLF11*
3	126.53–136.57	126,553,254	1	1.13 × 10^−6^	4.53 × 10^−1^	4.10 × 10^−5^	5.05 × 10^−5^	*TAF1B*
3	126.55–126.60	126,576,921	4	5.17 × 10^−7^	1.70 × 10^−1^	3.46 × 10^−6^	4.39 × 10^−6^	*TAF1B*
3	126.57–126.63	126,610,508	4	5.74 × 10^−6^	5.69 × 10^−2^	3.90 × 10^−6^	4.13 × 10^−6^	*TAF1B*
4	123.49–123.53	123,514,051	1	1.04 × 10^−5^	-	1.48 × 10^−6^	2.83 × 10^−6^	*BCAR3*
4	123.48–123.54	123,515,536	5	3.21 × 10^−5^	7.62 × 10^−1^	1.48 × 10^−6^	6.79 × 10^−6^	*BCAR3*
4	123.69–123.86	123,842,741	5	4.68 × 10^−6^	7.62 × 10^−1^	5.02 × 10^−4^	1.81 × 10^−3^	*FNBP1L*, *DR1*
4	124.53–124.57	124,546,985	3	2.58 × 10^−6^	-	7.72 × 10^−6^	2.58 × 10^−6^	*EVI5*
4	124.63–124.67	124,647,920	1	4.54 × 10^−6^	1.26 × 10^−1^	7.50 × 10^−2^	2.25 × 10^−2^	*GFI1*
4	124.88–124.93	124,907,426	136	4.62 × 10^−6^	-	2.15 × 10^−5^	4.62 × 10^−6^	*BTBD8*
4	124.84–124.93	124,912,144	83	2.38 × 10^−6^	6.77 × 10^−1^	4.03 × 10^−5^	2.03 × 10^−4^	*BTBD8*
4	124.89–124.95	124,934,706	71	1.67 × 10^−6^	3.45 × 10^−1^	2.50 × 10^−5^	3.62 × 10^−5^	*BTBD8*
4	124.93–124.97	124,946,497	7	2.48 × 10^−8^	5.10 × 10^−1^	1.05 × 10^−6^	6.64 × 10^−6^	*BTBD8*, *EPHX4*
4	124.92–125.00	124,977,862	102	1.55 × 10^−6^	8.97 × 10^−1^	5.50 × 10^−4^	6.65 × 10^−4^	*BTBD8*, *EPHX4*
4	124.96–125.00	124,978,259	3	2.88 × 10^−6^	-	2.79 × 10^−6^	2.88 × 10^−6^	*EPHX4*
4	125.08–125.29	125,266,938	27	2.59 × 10^−3^	9.16 × 10^−4^	3.54 × 10^−6^	7.94 × 10^−6^	** *TGFBR3* **
4	125.25–125.29	125,266,999	1	3.98 × 10^−6^	9.16 × 10^−4^	1.28 × 10^−8^	1.81 × 10^−8^	** *TGFBR3* **
4	125.25–125.33	125,314,460	38	1.39 × 10^−3^	9.61 × 10^−4^	2.35 × 10^−6^	4.41 × 10^−6^	** *TGFBR3* **
6	10.13–10.21	10,185,432	61	2.66 × 10^−6^	9.08 × 10^−1^	8.04 × 10^−2^	8.57 × 10^−4^	*NUDT7*
6	23.63–24.55	24,525,703	151	6.56 × 10^−1^	5.10 × 10^−8^	2.09 × 10^−3^	4.81 × 10^−5^	-
8	135.32–135.43	135,411,117	3	2.64 × 10^−6^	5.19 × 10^−1^	5.40 × 10^−3^	2.97 × 10^−3^	*LIN54*, *THAP9*, *SEC31A*
10	17.51–17.56	17,535,810	2	-	1.29 × 10^−3^	1.05 × 10^−6^	1.29 × 10^−3^	-
11	70.38–70.93	70,913,213	48	4.86 × 10^−3^	5.62 × 10^−5^	3.14 × 10^−6^	1.48 × 10^−6^	*FGF14*
11	70.89–70.94	70,915,607	13	8.02 × 10^−3^	5.62 × 10^−5^	8.92 × 10^−6^	2.68 × 10^−6^	-
14	31.82–31.86	31,835,585	1	5.00 × 10^−2^	6.43 × 10^−5^	4.63 × 10^−6^	3.06 × 10^−5^	*ARPC3*, *GPN3*, *FAM216A*
15	2.24–3.42	3,396,362	8	2.72 × 10^−4^	3.87 × 10^−3^	2.77 × 10^−6^	3.65 × 10^−6^	-
15	3.39–3.43	3,411,573	1	2.08 × 10^−1^	1.14 × 10^−6^	3.24 × 10^−6^	2.06 × 10^−5^	*EPC2*
15	3.53–3.57	3,545,949	1	2.09 × 10^−4^	1.65 × 10^−3^	1.79 × 10^−7^	1.20 × 10^−6^	*EPC2*
18	1.70–1.74	1,721,220	1	3.82 × 10^−3^	5.29 × 10^−5^	4.73 × 10^−6^	1.07 × 10^−6^	*MNX1*
18	1.69–1.78	1,758,722	4	4.17 × 10^−3^	2.37 × 10^−4^	2.26 × 10^−5^	4.09 × 10^−6^	*MNX1, HOM1*
18	4.16–5.70	5,675,281	71	1.88 × 10^−6^	1.12 × 10^−1^	1.94 × 10^−2^	1.77 × 10^−2^	*PRKAG2*
18	5.68–5.84	5,822,905	19	6.15 × 10^−6^	-	4.08 × 10^−6^	6.15 × 10^−6^	*PRKAG2*, *RHEB*, *CRYGN*
18	6.15–6.21	6,190,743	63	2.31 × 10^−6^	-	8.90 × 10^−7^	2.31 × 10^−6^	*AGAP3, FASTK, SLC4A2, ASIC3, ABCB8, ATG9B, NOS3*
18	6.18–6.25	6,231,446	52	1.16 × 10^−6^	-	2.01 × 10^−6^	1.16 × 10^−6^	*ASIC3, ABCB8, ATG9B, NOS3, KCNH2*
18	6.21–6.25	6,232,825	4	3.93 × 10^−6^	-	5.27 × 10^−6^	3.93 × 10^−6^	*NOS3, KCNH2*
18	6.21–6.27	6,246,190	25	3.93 × 10^−6^	-	1.32 × 10^−6^	3.93 × 10^−6^	*NOS3, KCNH2*
18	6.23–7.09	7,071,468	10	2.47 × 10^−6^	7.24 × 10^−1^	3.10 × 10^−4^	2.46 × 10^−4^	*KCNH2, GIMAP2, TAS2R39*
18	9.84–9.98	9,961,849	7	1.34 × 10^−4^	1.43 × 10^−2^	4.47 × 10^−6^	8.26 × 10^−6^	*TBXAS1, HIPK2*

SSC, *Sus scrofa* chromosome. SNP_R, range of significant and suggestive SNPs region. SNP_N, number of significant and suggestive SNPs. T_SNP_P, the position (bp) of the top SNP in range of significant and suggestive SNPs region. *p* (Canadian), *p*-value from within-population GWAS in the Canadian line. *p* (French), *p*-value from within-population GWAS in the French line. *p* (Combined LW), *p*-value from within-population GWAS in the combined lines of Large White pigs. *p* (Meta), *p*-value from cross-population meta-analyses. The bolded text shows the potential candidate gene detected in both chip and WGS data.

**Table 5 ijms-23-13338-t005:** The genome significant and suggestive SNPs for the gestation length (GL) trait using imputed WGS data in pigs.

SSC	SNP_R (Mb)	T_SNP_P (bp)	SNP_N	*p* (Canadian)	*p* (French)	*p* (Combined LW)	*p* (Meta)	Candidate Gene
1	254.70–254.77	254,753,857	15	9.06 × 10^−2^	2.19 × 10^−5^	4.87 × 10^−6^	3.40 × 10^−5^	*AMBP, KIF12, COL27A1*
1	254.73–254.78	254,756,216	10	6.79 × 10^−2^	7.84 × 10^−6^	1.31 × 10^−6^	4.91 × 10^−6^	*COL27A1*
1	254.74–254.78	254,757,687	5	3.92 × 10^−2^	3.23 × 10^−6^	7.39 × 10^−7^	1.77 × 10^−6^	*COL27A1*
2	148.22–148.26	148,238,122	1	6.20 × 10^−1^	3.91 × 10^−6^	4.08 × 10^−4^	4.16 × 10^−4^	*PPP2R2B*
6	156.54–156.58	156,564,691	2	5.73 × 10^−4^	8.95 × 10^−4^	1.06 × 10^−5^	1.71 × 10^−6^	-
6	156.55–156.60	156,578,616	2	3.48 × 10^−6^	6.73 × 10^−2^	2.07 × 10^−1^	3.43 × 10^−2^	-
6	159.28–159.49	159,467,436	4	3.98 × 10^−3^	5.68 × 10^−4^	3.85 × 10^−6^	2.45 × 10^−6^	*ZYG11B, PODN*
10	54.75–54.79	54,772,721	1	7.92 × 10^−3^	7.50 × 10^−4^	4.94 × 10^−6^	2.19 × 10^−5^	*MALRD1*
11	24.70–24.78	24,756,403	2	-	5.31 × 10^−7^	3.65 × 10^−6^	5.31 × 10^−7^	*AKAP11*
11	24.73–24.78	24,763,839	30	-	3.39 × 10^−7^	5.11 × 10^−5^	1.05 × 10^−5^	*AKAP11*
11	24.75–24.87	24,848,044	15	-	6.15 × 10^−7^	1.90 × 10^−3^	1.64 × 10^−3^	*DGKH*
11	24.93–25.02	24,996,968	3	-	3.98 × 10^−7^	6.50 × 10^−8^	3.98 × 10^−7^	*DGKH*
11	25.04–25.09	25,069,836	59	-	2.85 × 10^−6^	2.19 × 10^−4^	1.09 × 10^−4^	*VWA8*
11	25.05–25.09	25,072,777	5	-	5.46 × 10^−7^	8.49 × 10^−8^	5.46 × 10^−7^	*VWA8*
11	25.08–25.25	25,229,257	10	-	1.72 × 10^−6^	1.71 × 10^−5^	1.72 × 10^−6^	*VWA8*
11	25.21–25.33	25,306,498	58	-	8.50 × 10^−6^	3.40 × 10^−6^	8.50 × 10^−6^	*VWA8*
11	25.30–25.37	51,268,401	6	-	2.57 × 10^−6^	1.87 × 10^−7^	2.57 × 10^−6^	*VWA8*
11	51.25–51.29	51,268,401	1	2.87 × 10^−6^	4.07 × 10^−1^	6.18 × 10^−2^	4.61 × 10^−3^	-
14	61.85–62.44	62,420,628	46	2.87 × 10^−2^	2.29 × 10^−4^	3.67 × 10^−6^	2.30 × 10^−4^	** *BICC1* **
17	46.30–46.38	46,358,876	4	4.03 × 10^−6^	9.01 × 10^−1^	5.28 × 10^−3^	5.89 × 10^−4^	*GTSF1L*
18	45.35–45.39	45,371,853	1	4.60 × 10^−4^	1.55 × 10^−3^	3.95 × 10^−6^	2.36 × 10^−6^	*HOXA13*, *HOXA11*

SSC, *Sus scrofa* chromosome. SNP_R, range of significant and suggestive SNPs region. SNP_N, number of significant and suggestive SNPs. T_SNP_P, the position (bp) of the top SNP in range of significant and suggestive SNPs region. *p* (Canadian), *p*-value from within-population GWAS in the Canadian line. *p* (French), *p*-value from within-population GWAS in the French line. *p* (Combined LW), *p*-value from within-population GWAS in the both lines of Large White pigs. *p* (Meta), *p*-value from cross-population meta-analyses. The bolded text shows the potential candidate genes detected in both chip and WGS data.

## Data Availability

The datasets analyzed during this study are available from the authors upon reasonable request.

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
