# Peer review of "GWAS of Reproductive Traits in Large White Pigs on Chip and Imputed Whole-Genome Sequencing Data"

_ijms, 2022, doi:10.3390/ijms232113338_

Round 1

Reviewer 1 Report

Wang et al. performed multiple GWAS on reproductive traits in Large Whit pigs using SNP chip data and imputed whole-genome sequence data. My major concern about this study is, that the authors used breeding values for the traits. It is quite common to use de-regressed breeding values for GWAS in livestock, however it seems that the authors were not aware of  this procedure. That said, it did not make sense to evaluate the results of the study, as they could be completely different using de-regressed breeding values for GWAS. Therefore, I would like to encourage the authors to repeat the analyses using de-regressed breeding values.

Author Response

Thank you for taking the time to process the submission of our manuscript. We appreciate your valuable comments.

According to your comment, we calculated the DRP according to the method proposed by Garrick et al. (Deregressing estimated breeding values and weighting information for genomic regression analyses. Genet Sel Evol, 2009, 41, 55.). Just like Su et al. (Comparison of genomic predictions using genomic relationship matrices built with different weighting factors to account for locus specific variances. Journal of Dairy Science, 2014, 10:6547-6559), we deleted the deregressed proofs with reliability less than 10%. Hence, 20-200 individuals would be deleted in different traits. But, using these data, we didn’t identified any significant SNPs at the genome-wide level.

This maybe the crucial reason is that DRPs were calculated using ebvs, which can cause estimate error. Furthermore, Garrick et al. (2009) showed that DRPs had adjusted for parental average effect, but could still lead to higher FPRs when the EBVs were the results of repeated measurements (Ekine et al. Why breeding values estimated using familial data should not be used for genome-wide association studies. G3, 2014, 4:341-345).

In this study, we used the ebv+residual as adjusted phenotype to carry out GWAS for reproductive traits in Large White pigs. This idea comes from the paper published by Ekine et al. (Why breeding values estimated using familial data should not be used for genome-wide association studies. G3, 2014, 4:341-345). Finally, we still presented our GWAS results using the ebv+residual as phenotype. In revision, we added these information before the analysis model.

Reviewer 2 Report

Wang X. et al attempted to analyze and present results of a GWAS study on porcine economically important reproductive traits with success. Although the currently available porcine whole genome chips are less dense than those used for human studies, their results and in particular their study design, may be extended and be used in further studies in other mammals. 

Author Response

Thanks very much for your precious time to review this manuscript. I really appreciate your valuable comments. Thank you once again for your attention to our manuscript.

Reviewer 3 Report

Wang and coworkers described the identification of genetic variants, associated with reproductive traits in Large White pigs, by GWAS and cross-population meta-analyses. They identified several potential candidate genes for each of the economically important traits considered, namely the total number born, the number of stillborn and the gestation length.

The manuscript is clear and well written. The experimental procedures are well described, results are very convincing, and the conclusions are well supported by the described data. Even if the size of the population used is not very large, and detection power of GWAS analysis is not strong, results can be considered interesting.

Figures are of low quality and needs to be replaced by higher resolution images.

In my opinion, after image enhancement, the article can be published in IJMS.

Author Response

Thanks very much for taking your time to review this manuscript. I really appreciate your valuable and encouraging comments. We have considered these comments carefully and tried our best to address every one of them. In the future, we will expend the size of the population to improve the detection power of GWAS analysis. The figures are of low quality have been replaced by higher resolution images in the revision. We would like to thank you again for taking the time to review our manuscript.